# Less is More:
# Selective Layer Finetuning with SubTuning

## Abstract

Finetuning a pretrained model has become the standard approach for training neural networks on novel tasks, leading to rapid convergence and enhanced performance. In this work, we present a parameter-efficient finetuning method, wherein we selectively train a carefully chosen subset of layers while keeping the remaining weights frozen at their initial (pre-trained) values. We observe that not all layers are created equal: different layers across the network contribute variably to the overall performance, and the optimal choice of layers is contingent upon the downstream task and the underlying data distribution. We demonstrate that our proposed method, termed *subset finetuning* (or SubTuning), offers several advantages over conventional finetuning. We show that SubTuning outperforms both finetuning and linear probing in scenarios with scarce or corrupted data, achieving state-of-the-art results compared to competing methods for finetuning on small datasets. When data is abundant, SubTuning often attains performance comparable to finetuning while simultaneously enabling efficient inference in a multi-task setting when deployed alongside other models. We showcase the efficacy of SubTuning across various tasks, diverse network architectures and pre-training methods.

## 1 Introduction

Transfer learning from a large pretrained model has become a widely used method for achieving optimal performance on a diverse range of machine learning tasks in both Computer Vision and Natural Language Processing (Brown et al., 2020; Chowdhery et al., 2022; Wang et al., 2022; Yu et al., 2022). Traditionally, neural networks are trained "from scratch", where at the beginning of the training the weights of the network are randomly initialized. In transfer learning, however, we use the weights of a model that was already trained on a different task as the starting point for training on the new task, instead of using random initialization. In this approach, we typically replace the final (readout) layer of the model by a new "head" adapted for the new task, and tune the rest of the model (the backbone), starting from the pretrained weights. The use of a pretrained backbone allows leveraging the knowledge acquired from a large dataset, resulting in faster convergence time and improved performance, particularly when training data for the new downstream task is scarce.

The most common approaches for transfer learning are *linear probing* and *finetuning*. In linear probing, only the linear readout head is trained on the new task, while the weights of all other layers in the model are frozen at their initial (pretrained) values. This method is very fast and efficient in terms of the number of parameters trained, but it can be suboptimal due to its low capacity to fit the model to the new training data. Alternatively, it is also common to finetune all the parameters of the pretrained model to the new task. This method typically achieves better performance than linear probing, but it is often more costly in terms of training data and compute.

In this paper, we propose a simple alternative method, which serves as a middle ground between linear probing and full finetuning. Simply put, we suggest to train a carefully chosen small *subset* of layers in the network. This method, which we call SubTuning (see Figure 1), allows finding an optimal point between linear probing and full finetuning. SubTuning enjoys the best of both worlds: it is efficient in terms of the number of trained parameters, while still leveraging the computational capacity of training layers deep in the network. We show that SubTuning is a preferable transfer learning method when data is limited (see Figure 1), corrupted or in a multi-task setting with computational constraints. We compare our method in various settings to linear probing and finetuning, as well as other recent methods for Parameter-Efficient Transfer Learning (PETL) (e.g.,

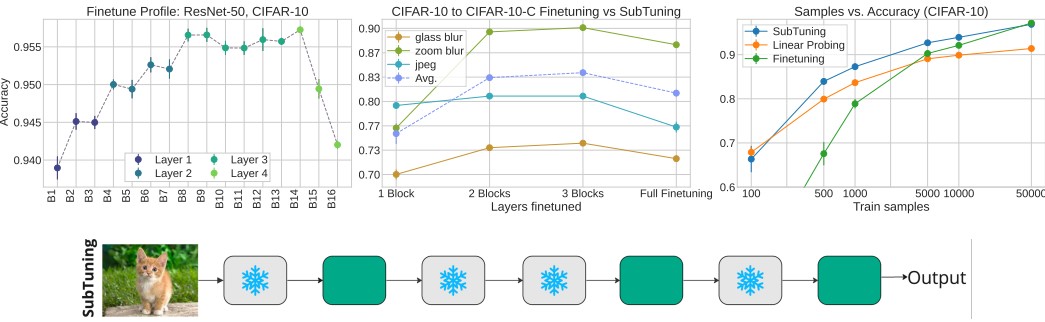

Figure 1: **Left.** The *Finetuning Profile* of ResNet-50 pretrained on ImageNet and finetuned on CIFAR-10. On the x-axis we have 16 res-blocks where each Layer (with Capital L) corresponds to a drop in spatial resolution. **Middle.** SubTuning on CIFAR-10-C distribution shifts with a ResNet-26. Even with few appropriately chosen residual blocks, SubTuning can be better than Finetuning. **Right.** Effect of dataset size on SubTuning, Finetuning and Linear Probing. SubTuning exhibits good performance across all dataset sizes, showcasing its flexibility. **Bottom.** *SubTuning illustration*. We only finetune a strategically selected subset of layers and the final readout layer, while the rest of the layers are frozen in their pretrained values.

Head2Toe (Evci et al., 2022) and LoRA (Hu et al., 2021)). Admittedly, our method incurs a high (but not prohibitive) training cost when selecting layers. However, we note that, unlike many PETL methods, our goal is to optimize for *deployment time* (inference time and algorithmic performance) rather than improving training efficiency.

**Our Contributions.** We summarize our contributions as follows:

- We advance our understanding of finetuning by introducing the concept of the *finetuning profile*, a valuable tool that sheds new light on the importance of individual layers during the finetuning process. This concept is further elaborated in Section 2.

- We present *SubTuning*, a simple yet effective algorithm that selectively finetunes specific layers based on a greedy selection strategy using the *finetuning profile*. In Section 3, we provide evidence that SubTuning frequently surpasses the performance of competing transfer learning methods in various tasks involving limited or corrupted data.

- We showcase the efficacy and computational run-time efficiency of SubTuning in the context of multi-task learning. This approach enables the deployment of multiple networks finetuned for distinct downstream tasks with minimal computational overhead, as discussed in Section 4.

## 1.1 RELATED WORK

**Parameter-Efficient Transfer-Learning.** In recent years, it became increasingly popular to fine-tune large pretrained models (Devlin et al., 2018; Radford et al., 2021; He et al., 2021b). As the popularity of finetuning these models grows, so does the importance of deploying them efficiently for solving new downstream tasks. Thus, there has been a growing interest, especially in the NLP domain, in Parameter-Efficient Transfer-Learning (PETL) (Rusu et al., 2016a; Sung et al., 2022; Evci et al., 2022; Zhang et al., 2019; Radiya-Dixit & Wang, 2020; Mallya & Lazebnik, 2018; Zaken et al., 2021; He et al., 2022) where we either modify a small number of parameters, add a few small layers or mask (Zhao et al., 2020) most of the network. Using only a fraction of the parameters for each task can help in avoiding catastrophic forgetting (McCloskey & Cohen, 1989) and can be an effective solution for both multi-task learning and continual learning. These methods encompass Prompt Tuning (Li & Liang, 2021; Lester et al., 2021; Jia et al., 2022), adapters (Houlsby et al., 2019; Rebuffi et al., 2017a; Chen et al., 2022; Rebuffi et al., 2018), LoRA (Hu et al., 2021), sidetun-ing (Zhang et al., 2019), feature selection (Evci et al., 2022) and masking (Radiya-Dixit & Wang, 2020). Fu et al. (2022), and He et al. (2021a), (see also references therein) attempt to construct a unified approach of PETL and propose improved methods.

In a recent study, Lee et al. (2022a), investigated the impact of selective layer finetuning on small datasets and found it to be more effective than traditional finetuning. They observed that the training

of different layers yielded varied results, depending on the shifts in data distributions. Specifically, they found that when there was a label shift between the source and target data, later layers performed better, but in cases of image corruption, early layers were more effective.

While our work shares some similarities with Lee et al., the motivations and experimental settings are fundamentally different. Primarily, we delve deeper into the complex interaction between the appropriate layers to finetune and the downstream task, pretraining objective, and model architecture, and observe that a more nuanced viewpoint is required. As evidenced by our *finetuning profiles* (e.g., see Figure 2 and Figure 5), a simple explanation of which layers to finetune based on the type of corruption is highly non-universal and the correct approach necessitates strategic layer selection, as demonstrated in our greedy method.

Moreover, we show that finetuning with layer selection is viable not only for adaptation to small corrupted data but also for general distribution shifts (in some of which we achieve state-of-the-art performance) and even for larger datasets. Additionally, our approach can be optimized for inference time efficiency in the Multi-Task Learning (MTL) setting.

We note that our *finetuning profiles* offer a unique insight into the mechanistic understanding of finetuning, making our research not only practical in the MTL and PETL settings but also scientifically illuminating. We also note that SubTuning is compatible with many other PETL methods, and composing SubTuning with methods like LoRA and Head2Toe is a promising research direction that we leave for future work.

**Multi-Task Learning.** Neural networks are often used for solving multiple tasks. These tasks typically share similar properties, and solving them concurrently allows sharing common features that may capture knowledge that is relevant for all tasks (Caruana, 1997). However, MTL also presents significant challenges, such as negative transfer (Liu et al., 2019), loss balancing (Lin et al., 2022; Michel et al., 2022), optimization difficulty (Pascal et al., 2021), data balancing and shuffling (Purushwalkam et al., 2022). While these problems can be mitigated by careful sampling of the data and tuning of the loss function, these solutions are often fragile (Worsham & Kalita, 2020). In a related setting called *Continual Learning* (Rusu et al., 2016b; Rebuffi et al., 2017b; Kirkpatrick et al., 2017; Kang et al., 2022), adding new tasks needs to happen on-top of previously deployed tasks, while losing access to older data due to storage or privacy constraints, complicating matters even further. In this context, we show that new tasks can be efficiently added using SubTuning, without compromising performance or causing degradation of previously learned tasks (Section 4).

## 2 NOT ALL LAYERS ARE CREATED EQUAL

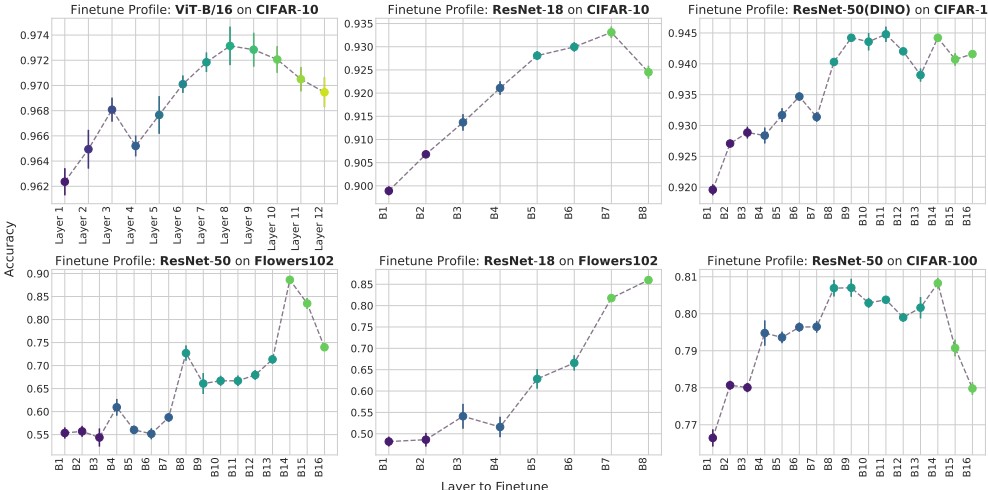

Figure 2: Finetuning profiles for different architectures, initializations and datasets.

In the process of finetuning deep neural networks, a crucial yet often undervalued aspect is the unequal contribution of individual layers to the model's overall performance. This variation in layer

importance calls into question prevalent assumptions and requires a more sophisticated approach to effectively enhance the finetuning process. By selectively training layers, it is possible to strategically allocate computational resources and improve the model's performance. To pinpoint the essential components within the network, we examine two related methods: constructing the *finetuning profile* by scanning for the optimal layer (or block of layers) with a complexity of $O(\text{num layers})$, and a Greedy SubTuning algorithm, where we iteratively leverage the finetuning profile to select $k$-layers one by one, while using a higher complexity of $O(\text{num layers} \cdot k)$.

**The Finetuning Profile.** We commence by conducting a comprehensive analysis of the significance of finetuning different components of the network. This analysis guides the choice of the subset of layers to be used for SubTuning. To accomplish this, we run a series of experiments in which we fix a specific subset of consecutive layers within the network and finetune only these layers, while maintaining the initial (pretrained) weights for the remaining layers.

For example, we take a ResNet-50 pretrained on the ImageNet dataset, and finetune it on the CIFAR-10 dataset, replacing the readout layer of ImageNet (which has 1000 classes) by a readout layer adapted to CIFAR-10 (with 10 classes). As noted, in our experiments we do not finetune all the weights of the network, but rather optimize only a few layers from the model (as well as the readout layer). Specifically, as the ResNet-50 architecture is composed of 16 blocks (i.e., *ResBlocks*, see Appendix A and He et al. (2015) for more details), we choose to run 16 experiments, where in each experiment we train only one block, fixing the weights of all other blocks at their initial (pretrained) values. We then plot the accuracy of the model as a function of the block that we train, as presented in Figure 1 left. We call this graph the *finetuning profile* of the network. Following a similar protocol (see Appendix A), we compute *finetuning profiles* for various combinations of architectures (ResNet-18, ResNet-50 and ViT-B/16), pretraining methods (supervised and DINO (Caron et al., 2021)), and target tasks (CIFAR-10, CIFAR-100 and Flower102). In Figure 2, we present the profiles for the different settings.

**Results.** Interestingly, our findings indicate that for most architectures and datasets, the importance of a layer cannot be predicted by simply observing properties such as the depth of the layer, the number of parameters in the layer or its spatial resolution. In fact, the same architecture can have distinctively different *finetuning profiles* when trained on a different downstream task or from different initialization (see Figure 2). While we find that layers closer to the input tend to contribute less to the finetuning process, the performance of the network typically does not increase monotonically with the depth or with the number of parameters[1], and after a certain point the performance often starts *decreasing* when training deeper layers. For example, in the finetuning profile of ResNet-50 finetuned on the CIFAR-10 dataset (Figure 1 left), we see that finetuning Block 13 results in significantly better performance compared to optimizing Block 16, which is deeper and has many more parameters. We also look into the effect of finetuning more consecutive blocks. In Fig-

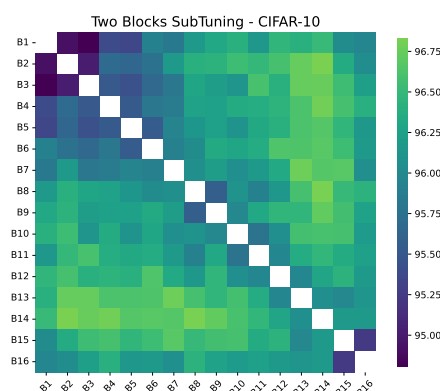

Figure 3: 2-block finetuning profile for ResNet-50 over CIFAR-10.

ure 9 (in Appendix B) we present the finetuning profiles for training groups of 2 and 3 consecutive blocks. The results indicate that finetuning more layers improves performance, and also makes the finetuning profile more monotonic.

**Greedy Selection.** The discussion thus far prompts an inquiry into the consequences of training arbitrary (possibly non-consecutive) layers. First, we observe that different combinations of layers admit non-trivial interactions, and therefore simply choosing subsets of consecutive layers may be suboptimal. For example, in Figure 3 we plot the accuracy of training all possible subsets of two blocks from ResNet-50, and observe that the optimal performance is achieved by Block 2 and Block 14. Therefore, a careful selection of layers to train is needed.

---

[1]In the ResNet architectures, deeper blocks have more parameters, while for ViT all layers have the same amount of parameters.

A brute-force approach for testing all possible subsets of $k$ layers would result in a computational burden of $O(\text{num layers}^k)$. To circumvent this issue, we introduce an efficient greedy algorithm with a cost of $O(\text{num layers} \cdot k)$. This algorithm iteratively selects the layer that yields the largest marginal contribution to validation accuracy, given the currently selected layers. The layer selection process is halted when the marginal benefit falls below a predetermined threshold, $\varepsilon$, after which the chosen layers are finetuned. The pseudo-code for this algorithm is delineated in Algorithm 1 in Appendix A. We note that such greedy optimization is a common approach for subset selection in various combinatorial problems, and is known to approximate the optimal solution under certain assumptions. We show that SubTuning results in comparable performance to full finetuning even for full datasets (see Figure 1 right).

## 2.1 THEORETICAL MOTIVATION

We now provide some theoretical justification for using Greedy SubTuning when data size is limited. Denote by $\theta \in \mathbb{R}^r$ an initial set of pretrained parameters, and by $f_\theta$ the original network that uses these parameters. In standard finetuning, we tune $\theta$ on the new task, resulting in some new set of parameters $\widetilde{\theta}$, satisfying $\left\| \widetilde{\theta} - \theta \right\|_2 \leq \Delta$. Using first-order taylor expansion, when $\Delta$ is small, we get:

$$f_{\widetilde{\theta}}(\mathbf{x}) \approx f_\theta(\mathbf{x}) + \left\langle \nabla f_\theta(\mathbf{x}), \widetilde{\theta} - \theta \right\rangle = \langle \psi_\theta(\mathbf{x}), \mathbf{w} \rangle$$

for some mapping of the input $\psi_\theta$ (typically referred to as the Neural Tangent Kernel (Jacot et al., 2018)), and some vector $\mathbf{w}$ of norm $\leq \Delta$. Now, if we optimize $\mathbf{w}$ over some dataset of size $m$, using standard norm-based generalization bounds (Shalev-Shwartz & Ben-David, 2014), we can show that the generalization of the resulting classifier is $O\left(\frac{\sqrt{r}\Delta}{\sqrt{m}}\right)$, where $r$ is the number of parameters. Thus, when the number of parameters is large, many samples will be required for good performance.

SubTuning can potentially lead to much better generalization guarantees. Since in SubTuning we train only a subset of the network's parameters, we could hope that the generalization depends only on the number of parameters in the trained layers. This is not immediately true, since the Greedy SubTuning algorithm reuses the same dataset while searching for the optimal subset, which can potentially increase the sample complexity (i.e., when the optimal subset is "overfitted" to the training set). However, a careful analysis reveals that the Greedy SubTuning indeed allows improved guarantees, and that the subset selection only adds logarithmic factors to sample complexity:

**Theorem 1.** If we run *Greedy SubTuning* over a network with $L$ layers, tuning at most $k$ layers with $r' \ll r$ parameters. Then the generalization error of the resulting classifier is $O\left(\frac{\sqrt{r'}\Delta \log(kL)}{\sqrt{m}}\right)$.

We give the proof of the above theorem in the Appendix. Observe that the experiments reported in Figure 1 (right) indeed validate the superiority of SubTuning in terms of sample complexity.

## 3 SUBTUNING FOR LOW DATA REGIME

In this section, we focus on finetuning in the low-data regime. As mentioned, transfer learning is a common approach in this setting, leveraging the power of a model that is already pretrained on large amounts of data. We show that in this context, SubTuning can outperform both linear probing and full finetuning, as well as other parameter efficient transfer learning methods. Additionally, we demonstrate the benefit of using SubTuning when data is corrupted.

### 3.1 EVALUATING SUBTUNING IN LOW-DATA REGIMES

We study the advantages of using SubTuning when data is scarce, compared to other transfer learning methods. Beside linear probing and finetuning, we also compare our method to highly performing algorithms in the low data regime: Head2Toe Evci et al. (2022) and LoRA Hu et al. (2021). Head2Toe is a method for bridging the gap between linear probing and finetuning, which operates by training a linear layer on top of features selected from activation maps throughout the network. LoRA is a method that trains a "residual" branch (mostly inside a Transformer) using a low rank decomposition of the layer.

Table 1: Performance of ResNet-50 and ViT-b/16 pretrained on ImageNet and finetuned on datasets from VTAB-1k. We only present results for surgically fine-tuning the last layer, layer 4, as it outperformed all other layers on all datasets. We point out that surgical fine-tuning isn't suitable for ViT-b/16 due to its lack of defined block groupings into layers, unlike ResNet-50.

| | ResNet50 | | | | ViT-b/16 | | | |
| | CIFAR-100 | Flowers102 | Caltech101 | DMLAB | CIFAR-100 | Flowers102 | Caltech101 | DMLab |
| --- | --- | --- | --- | --- | --- | --- | --- | --- |
| Ours | **54.6** | **90.5** | 86.5 | **51.2** | **68.0** | **97.7** | 86.5 | 36.4 |
| H2T | 47.1 | 85.6 | **88.8** | 43.9 | 58.2 | 85.9 | **87.3** | **41.6** |
| FT | 33.7 | 87.3 | 78.7 | 48.2 | 47.8 | 91.2 | 80.7 | 34.3 |
| LP | 35.4 | 64.2 | 67.1 | 36.3 | 29.9 | 84.7 | 72.7 | 31.0 |
| LoRA | - | - | - | - | 40.4 | 88.3 | 79.2 | 36.4 |
| Surgical FT | 47.3 | 90.1 | 84.4 | 46.8 | - | - | - | - |

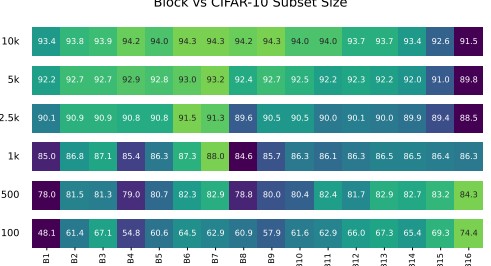

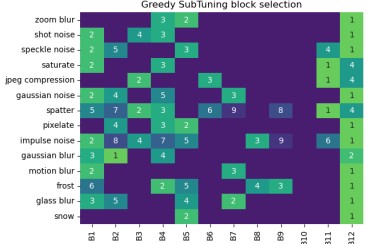

Figure 4: Single block SubTuning of ResNet-50 on CIFAR-10. The y axis is dataset size, x axis is the chosen block. With growing dataset sizes, training earlier layers proves to be more beneficial.

Figure 5: Block selection profiling for the Greedy SubTuning method, showing the order of block selection for each data corruption.

**VTAB-1k.** First, we evaluate the performance of SubTuning on the VTAB-1k benchmark, focusing on the CIFAR-100, Flowers102, Caltech101, and DMLab datasets using the 1k examples split specified in the protocol. We employed the Greedy SubTuning approach described in Section 2 to select the subset of layers to finetune. For layer selection, we divided the training dataset into five parts and performed five-fold cross-validation. We used the official PyTorch ResNet-50 pretrained on ImageNet and ViT-b/16 pretrained on ImageNet-22k from the official repository of Ridnik et al. (2021). The results are presented in Table 1. Our findings indicate that SubTuning frequently outperforms competing methods and remains competitive in other cases.

**Effect of Dataset Size.** The optimal layer selection for a given task is contingent upon various factors, such as the architecture, the task itself, and the dataset size. We proceed to investigate the impact of dataset size on the performance of SubTuning with different layers by comparing the finetuning of a single residual block to linear probing and finetuning on CIFAR-10 with varying dataset sizes. We present the results in Figure 4. Our findings demonstrate that layers closer to the output exhibit superior performance when training on smaller datasets.

In addition to these experiments, we also explore the use of SubTuning in a pool-based active learning (AL) setting, where a large pool of unlabeled data is available, and additional examples can be labeled to improve the model's accuracy. Our results suggest that SubTuning outperforms both linear probing and full finetuning in this setting. See our results in Figure 11 in the Appendix.

## 3.2 DISTRIBUTION SHIFT AND DATA CORRUPTION

Deep neural networks are known to be sensitive to minor distribution shifts between the source and target domains, which lead to a decrease in their performance (Recht et al., 2019; Hendrycks & Dietterich, 2019; Koh et al., 2021). One cost-effective solution to this problem is to collect a small labeled dataset from the target domain and finetune a pretrained model on this dataset. In this section, we focus on a scenario where a large labeled dataset is available from the source domain, but only limited labeled data is available from the target domain. We demonstrate that Greedy SubTuning

---

[1]Results from original paper. Pretrained models can differ due to difference in software suit (TensorFlow).

yields better results compared to finetuning all layers, and also compared to Surgical finetuning (Lee et al., 2022b), where a large subset of consecutive blocks is trained.

Table 2: CIFAR-10 to CIFAR-10-C distribution shift.

| Distribution shift | SubTuning | Finetuning | Surgical L1 | Surgical L2 | Surgical L3 | Linear |
|---|---|---|---|---|---|---|
| zoom blur | $\mathbf{90.0 \pm 0.1}$ | $87.8 \pm 0.4$ | $89.2 \pm 0.1$ | $89.1 \pm 0.2$ | $85.5 \pm 0.3$ | $68.7 \pm 0.04$ |
| speckle noise | $\mathbf{81.5 \pm 0.2}$ | $77.8 \pm 0.6$ | $78.4 \pm 0.1$ | $74.8 \pm 0.1$ | $71.1 \pm 0.1$ | $51.5 \pm 0.01$ |
| spatter | $89.2 \pm 0.2$ | $86.8 \pm 0.3$ | $\mathbf{89.4 \pm 0.1}$ | $87.4 \pm 0.2$ | $85.3 \pm 0.0$ | $80.4 \pm 0.07$ |
| snow | $\mathbf{86.0 \pm 0.2}$ | $84.1 \pm 0.2$ | $84.8 \pm 0.2$ | $84.3 \pm 0.1$ | $82.2 \pm 0.2$ | $78.7 \pm 0.07$ |
| shot noise | $\mathbf{82.0 \pm 0.3}$ | $77.6 \pm 0.4$ | $77.0 \pm 0.9$ | $74.2 \pm 0.1$ | $69.9 \pm 0.1$ | $46.4 \pm 0.01$ |
| saturate | $\mathbf{92.0 \pm 0.1}$ | $89.5 \pm 0.3$ | $91.7 \pm 0.0$ | $91.2 \pm 0.0$ | $90.4 \pm 0.0$ | $89.8 \pm 0.04$ |
| pixelate | $\mathbf{86.1 \pm 0.0}$ | $82.8 \pm 0.5$ | $85.8 \pm 0.1$ | $83.6 \pm 0.2$ | $78.5 \pm 0.2$ | $54.8 \pm 0.02$ |
| motion blur | $\mathbf{87.3 \pm 0.1}$ | $85.5 \pm 0.3$ | $86.7 \pm 0.1$ | $86.9 \pm 0.1$ | $83.4 \pm 0.1$ | $72.9 \pm 0.03$ |
| jpeg compression | $\mathbf{80.8 \pm 0.2}$ | $76.5 \pm 0.7$ | $80.1 \pm 0.5$ | $76.8 \pm 0.1$ | $74.9 \pm 0.1$ | $72.0 \pm 0.04$ |
| impulse noise | $\mathbf{75.4 \pm 0.5}$ | $70.8 \pm 0.7$ | $69.6 \pm 0.3$ | $63.8 \pm 0.1$ | $56.7 \pm 0.1$ | $35.2 \pm 0.01$ |
| glass blur | $\mathbf{74.3 \pm 0.3}$ | $72.2 \pm 0.2$ | $69.9 \pm 0.4$ | $71.5 \pm 0.1$ | $67.8 \pm 0.1$ | $55.2 \pm 0.06$ |
| gaussian noise | $\mathbf{80.0 \pm 0.2}$ | $75.1 \pm 1.2$ | $72.7 \pm 0.1$ | $71.0 \pm 0.1$ | $66.6 \pm 0.2$ | $41.1 \pm 0.01$ |
| gaussian blur | $\mathbf{89.5 \pm 0.2}$ | $86.4 \pm 0.4$ | $88.1 \pm 0.0$ | $87.3 \pm 0.1$ | $80.0 \pm 0.0$ | $41.7 \pm 0.05$ |
| frost | $84.2 \pm 0.2$ | $83.1 \pm 0.4$ | $\mathbf{84.2 \pm 0.3}$ | $83.2 \pm 0.1$ | $80.4 \pm 0.2$ | $68.5 \pm 0.03$ |
| Average | $\mathbf{84.2 \pm 0.2}$ | $81.1 \pm 0.5$ | $82.0 \pm 0.2$ | $80.4 \pm 0.1$ | $76.6 \pm 0.1$ | $61.2 \pm 0.04$ |

In this section, we follow the setup proposed by Lee et al. (2022b), analyzing the distribution shift from CIFAR-10 to CIFAR-10-C (Hendrycks & Dietterich, 2019) for ResNet-26. The task is to classify images where the target distribution is composed of images of the original distribution with added input corruption out of a predefined set of 14 corruptions. For details refer to Appendix A.

**Results.** In Table 2, we display the performance of linear probing, finetuning, Surgical finetuning[2] as well as SubTuning. We see that our method often outperforms and always is competitive with other methods. On average, SubTuning performs 3% better than full finetuning and 2.2% better than Surgical finetuning reproduced in our setting[3].

Next, we analyse the number of residual blocks required for SubTuning as shown in Figure 1(middle). We report the average accuracy on 3 distribution shifts (glass blur, zoom blur and jpeg compression) and the average prerformance for the 14 corruptions in CIFAR-10-C. Even with as little as 2 appropriately selected residual blocks, SubTuning shows better performance than full finetuning.

Finally, we analyze which blocks were used by the Greedy-SubTuning method above. Figure 5 illustrates the selected blocks and their respective order for each dataset. Our findings contradict the commonly held belief that only the last few blocks require adjustment. In fact, SubTuning utilizes numerous blocks from the beginning and middle of the network. Furthermore, our results challenge the claim made in Lee et al. (2022b) that suggests adjusting only the first layers of the network suffices for input-level shifts in CIFAR-10-C. Interestingly, we found that the ultimate or penultimate block was the first layer selected for all corruptions, resulting in the largest performance increase.

## 4 EFFICIENT MULTI-TASK LEARNING WITH SUBTUNING

So far, we demonstrated the varying impact of different layers on the overall performance of a finetuned model, showing that high accuracy can be achieved without training all parameters of the network, provided that the right layers are selected for training. In this section, we focus on utilizing SubTuning for Multi-Task Learning (MTL).

One major drawback of standard finetuning in the context of multi-task learning Caruana (1997); Ruder (2017) is that once the model is finetuned on a new task, its weights may no longer be suitable for the original source task (a problem known as *catastrophic forgetting* McCloskey & Cohen (1989)). Consider for instance the following multi-task setting, which serves as the primary motivation for this section. Assume we have a large backbone network that was trained on some source

---

[2]Surgical finetuning focuses on training whole Layers (which consist of 4 blocks for ResNet26).

[3]We note that (Lee et al., 2022b) reports slightly higher performance than our reproduction, but still achieves accuracy that is lower by 1.4% compared to SubTuning.

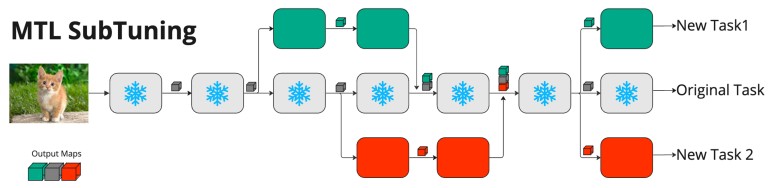

Figure 6: SubTuning for MTL. Each new task utilizes a consecutive subset of layers of a network and shares the others. At the end of the split, the outputs of different tasks are concatenated and parallelized along the batch axis for computational efficiency.

task, and is already deployed and running as part of our machine learning system. When presented with a new task, we finetune our deployed backbone on this task, and want to run the new finetuned network in parallel to the old one. This presents a problem, as we must now run the same architecture twice, each time with a different set of weights. Doing so doubles the cost both in terms of compute (the number of multiply-adds needed for computing both tasks), and in terms of memory and IO (the number of bits required to load the weights of both models from memory). An alternative would be to perform multi-task training for both the old and new task, but this usually results in degradation of performance on both tasks, with issues such as data balancing, parameters sharing and loss weighting cropping up Chen et al. (2018); Sun et al. (2020); Sener & Koltun (2018).

We show that using SubTuning, we can efficiently deploy new tasks at inference time (see Figure 6), with minimal cost in terms of compute, memory and IO, while maintaining high accuracy on the downstream tasks. Instead of training all tasks simultaneously, which can lead to task interference and complex optimization, we propose starting with a network pretrained on some primary task, and adding new tasks with SubTuning on top of it (Figure 6). This framework provides assurance that the performance of previously learned tasks will be preserved while adding new tasks.

## 4.1 COMPUTATIONALLY EFFICIENT INFERENCE

We will now demonstrate how SubTuning improves the computational efficiency of the network at *inference time*, which is the primary motivation for this section.

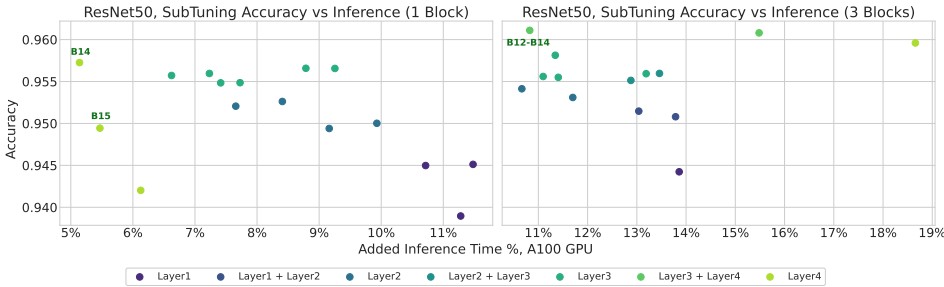

Figure 7: Accuracy on CIFAR-10 vs A100 latency with batch size of 1 and input resolution of 224.

Let us consider the following setting of multi-task learning. We trained a network $f_\theta$ on some task. The network gets an input $\mathbf{x}$ and returns an output $f_\theta(\mathbf{x})$. We now want to train a new network on a different task by finetuning the weights $\theta$, resulting in a new set of weights $\widetilde{\theta}$. Now, at inference time, we receive an input $\mathbf{x}$ and want to compute both $f_\theta(\mathbf{x})$ and $f_{\widetilde{\theta}}(\mathbf{x})$ with minimal compute budget. Since we cannot expect the overall compute to be lower than just running $f_\theta(\mathbf{x})$[4], we only measure the *additional* cost of computing $f_{\widetilde{\theta}}(\mathbf{x})$, given that $f_\theta(\mathbf{x})$ is already computed.

Since inference time heavily depends on various parameters such as the hardware used for inference (e.g., CPU, GPU, FPGA), the hardware parallel load, the network compilation (i.e., kernel fusion) and the batch size, we will conduct a crude analysis of the compute requirements (see in depth discussion in Li et al. (2021)). The two main factors that contribute to computation time are: 1)

---

[4]Optimizing the compute budget of a single network is outside the scope of this paper.

**Computational cost**, or the number of multiply-adds (FLOPs) needed to compute each layer and 2) **IO**, which refers to the number of bits required to read from memory to load each layer's weights.

If we perform full finetuning of all layers, in order to compute $f_{\widetilde{\theta}}(\mathbf{x})$ we need to double both the computational cost and the IO, as we are now effectively running two separate networks, $f_\theta$ and $f_{\widetilde{\theta}}$, with two separate sets of weights. Note that this does not necessarily mean that the computation-time is doubled, since most hardware used for inference does significant parallelization, and if the hardware is not fully utilized when running $f_\theta(\mathbf{x})$, the additional cost of running $f_{\widetilde{\theta}}(\mathbf{x})$ in parallel might be smaller. However, in terms of additional compute, full finetuning is the least optimal thing to do.

Consider the computational cost of SubTuning. For simplicity we analyze the case where the chosen layers are consecutive, but similar analysis can be applied to the non-consecutive case. Denote by $N$ the number of layers in the network, and assume that the parameters $\widetilde{\theta}$ differ from the original parameters $\theta$ only in the layers $\ell_{\text{start}}$ through $\ell_{\text{end}}$ (where $1 \leq \ell_{\text{start}} \leq \ell_{\text{end}} \leq N$). Let us separate between two cases: 1) $\ell_{\text{end}}$ is the final layer of the network and 2) $\ell_{\text{end}}$ is some intermediate layer.

The case where $\ell_{\text{end}}$ is the final layer is the simplest: we share the entire compute of $f_\theta(\mathbf{x})$ and $f_{\widetilde{\theta}}(\mathbf{x})$ up until the layer $\ell_{\text{start}}$ (so there is zero extra cost for layers below $\ell_{\text{start}}$), and then we "fork" the network and run the layers of $f_\theta$ and $f_{\widetilde{\theta}}$ in parallel. In this case, both the compute and the IO are doubled *only* for the layers between $\ell_{\text{start}}$ and $\ell_{\text{end}}$.

In the second case, where $\ell_{\text{end}}$ is some intermediate layer, the computational considerations are more nuanced. As in the previous case, we share the entire computation before layer $\ell_{\text{start}}$, with no extra compute. Then we "fork" the network, paying double compute and IO for the layers between $\ell_{\text{start}}$ and $\ell_{\text{end}}$. For the layers after $\ell_{\text{end}}$, however, we can "merge" back the outputs of the two parallel branches (i.e., by "batch" axis concatenation), and use the same network weights for both outputs. This means that for the layers after $\ell_{\text{end}}$ we double the compute (i.e., in FLOPs), but the IO remains the same (by reusing the weights for both outputs). This mechanism is illustrated in Figure 6.

More formally, let $c_i$ be the computational-cost of the $i$-th layer, and let $s_i$ be the IO required for the $i$-th layer. To get a rough estimate of how the IO and compute affect the backbone run-time, consider a simple setting where compute and IO are parallelized. Thus, while the processor computes layer $i$, the weights of layer $i + 1$ are loaded into memory. The total inference time of the model is then:

$$\text{Compute} = \max(2s_{\ell_{\text{start}}}, c_{\ell_{\text{start}}-1}) + \sum_{i=\ell_{\text{start}}}^{\ell_{\text{end}}} 2\max(c_i, s_{i+1}) + \sum_{i=\ell_{\text{end}}+1}^{N-1} \max(2c_i, s_{i+1}) + 2c_N$$

Thus, both deeper and shallower layers can be optimal for SubTuning, depending on the exact deployment environment, workload and whether we are IO or compute bound. We proceed to empirically investigate the performance vs latency tradeoffs of SubTuning for MTL. We conduct an experiment using ResNet-50 on an NVIDIA A100-SXM-80GB GPU with a batch size of 1 and resolution 224. We finetune 1 and 3 consecutive res-blocks and plot the accuracy against the added inference cost, as seen in Figure 7. This way we are able to achieve significant performance gains, with minimal computational cost. However, it is important to mention that the exact choice of which layer gives the optimal accuracy-latency tradeoff can heavily depend on the deployment environment, as the runtime estimation may vary depending on factors such as hardware, job load, and software stack. For further investigation of accuracy-latency in the MTL setting refer to Appendix B.

## 5 DISCUSSION

Neural networks are now becoming an integral part of software development. In conventional development, teams can work independently and resolve conflicts using version control systems. But with neural networks, maintaining independence becomes difficult. Teams building a single network for different tasks must coordinate training cycles, and changes in one task can impact others. We believe that SubTuning offers a viable solution to this problem. It allows developers to "fork" deployed networks and develop new tasks without interfering with other teams. This approach promotes independent development, knowledge sharing, and efficient deployment of new tasks. As we showed, it also results in improved performance compared to competing transfer learning methods in different settings. In conclusion, we hope that SubTuning, along with other efficient finetuning methods, may play a role in the ongoing evolution of software development in the neural network era.

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

## A    EXPERIMENTAL SETUP

Unless stated otherwise, for experiments throughout the paper we used a fixed experimental setting presented in Table 3. We focus on short training of 10 epochs, using the AdamW Loshchilov & Hutter (2017) optimizer on image resolution of 224 with random resized crop to transform from lower to larger resolution. We also employ random horizontal flip and for Flowers102 we use random rotation up to probability 30%. We make a few exceptions to this setting. One, in Section 3, due to scarce data, we train for 50 epochs. Also, in Subsection 3.2, where we train for 15 epochs and use the same learning rate tuning for layer selection as in Lee et al. (2022b). We report our results on the CIFAR-10, CIFAR-100 Krizhevsky et al. (2009), Flower102 Nilsback & Zisserman (2008) and Standford Cars Krause et al. (2013), in addition to the CIFAR-C results in reported in in Subsection 3.2.

Table 3: Training Parameters. For ViT-B/16 we use two sets of parameters. One for a full length datasets and the other for small datasets with 1k training examples (Table 1).

|  | ResNet | ViT-B/16 full datasets | ViT-B/16 1k examples |
|---|---|---|---|
| Learning Rate | 0.001* | 0.0001 | 0.001 |
| Weight Decay | 0.01 | 0.00005 | 0.01 |
| Batch Size | 256 | 256 | 256 |
| Optimizer | AdamW | AdamW | AdamW |
| Scheduler | Cosine Annealing | Cosine Annealing | Cosine Annealing |

*For linear probing with ResNet, we use learning rate of 0.01.

**Pretrained Weights**    For ResNet-50, ResNet-18 He et al. (2015) and ViT-B/16 Dosovitskiy et al. (2020) we use pretrained weights using the default TorchVision implementation Marcel & Rodriguez (2010) pretrained on ImageNet Deng et al. (2009). For the DINO ResNet-50 Caron et al. (2021), we used the official paper github's weights https://github.com/facebookresearch/dino.

**Finetuning Profiles.**    To generate the finetuning profiles we only train the appropriate subset of residual blocks (for ResNets) and Self-Attention layers (For ViT) in addition to training an appropriate linear head. For example, for ResNet-18, there are 8 residual blocks (ResBlocks), 2 in each layer or spatial resolution (see full implementation here: link.). In general, each such block consists of a few Convolution layers with a residual connection that links the input of the block and the output of the Conv-layers. Similarly, ResNet-50 has 16 blocks, [3, 4, 6, 3], for layers [1, 2, 3, 4] respectively. For ViT-B/16, there are naturally 12 attention layers and we train one (or few) layer at a time.

**Greedy SubTuning.**    We evaluated each subset of Blocks using 5-fold cross validation in all of our experiments where we used the greedy algorithm. In Algorithm 1 we present the pseudo code for Greedy SubTuning. Table 4 shows the Blocks selected by the greedy algorithm for CIFAR-10 subsets of different size, producing the results presented in Figure 1 (right).

Table 4: Selected Blocks of ResNet-50 for Different Training Set Sizes of CIFAR-10

| CIFAR-10 Subset Size | ResNet-50 Selected Blocks |
|---|---|
| 100 | 6, 11, 13, 16 |
| 500 | 8, 13, 15, 16 |
| 1,000 | 7, 9, 11, 15 |
| 5,000 | 4, 8, 11, 13, 15 |
| 10,000 | 3, 5, 8, 9, 10, 14, 15, 16 |
| 50,000 | 2, 4, 5, 12, 13, 14, 15, 16 |

---

**Algorithm 1** Greedy-SubTuning

---

1: **procedure** GREEDYSUBSETSELECTION(model, all_layers, $\varepsilon$)
2:     $S \leftarrow \{\}, n \leftarrow |\text{all\_layers}|$
3:     $A_{best} = 0$
4:     **for** $i = 1$ to $n$ **do**
5:         $A_{iter} \leftarrow 0, L_{best} \leftarrow$ null
6:         **for** $L \in (\text{all\_layers} - S)$ **do**
7:             $S' \leftarrow S \cup \{L\}$
8:             $A_{new} \leftarrow$ evaluate(model, $S'$)
9:             **if** $A_{new} > A_{iter}$ **then**
10:                 $L_{best} \leftarrow L, A_{iter} \leftarrow A_{new}$
11:             **end if**
12:         **end for**
13:         **if** $A_{iter} > A_{best} + \varepsilon$ **then**
14:             $A_{best} \leftarrow A_{iter}, S \leftarrow S \cup \{L_{best}\}$ # if no layers helps sufficiently, we stop
15:         **else**
16:             Break
17:         **end if**
18:     **end for**
19:     **return** $S$
20: **end procedure**

---

**Data corruption.** Throughout Section 3.2 we follow the setup proposed by Lee et al. (2022b), analyzing the distribution shift from CIFAR-10 to CIFAR-10-C Hendrycks & Dietterich (2019) for ResNet-26. The task is to classify images where the target distribution is composed of images of the original distribution with added input corruption out of a predefined set of 14 corruptions.

Similarly to Lee et al. (2022b), for each corruption we use 1k images as a train set and 9k as a test set. For the layer selection we perform 5 fold cross-validation using only the 1k examples of the training set, and only after the layers subset is selected we train on the full 1k training data, evaluating on the test set. We use the ResNet-26 model with "Standard" pretraining and data loading code from Croce et al. Croce et al. (2020). We use the highest corruption severity of 5. We tune over 5 learning rates 1e-3, 5e-4, 1e-4, 5e-5, 1e-5 and report the average of 5 runs.

**Inference Time Measurement.** We measure inference time on a single NVIDIA A100-SXM-80GB GPU with a batch size of 1 and input resolution 224. We warm up the GPU for 300 iteration and run 300 further iterations to measuring the run time. Since measuring inference time is inherently noisy, we make sure the number of other processes running on the GPU stays minimal and report the mean time out of 10 medians of 300. We attach figures for absolution times in Figure 18.

A.1 EXPERIMENTAL SETUP FOR ABLATIONS

**Pruning.** We use the Torch-Pruning library Fang (2022) to apply both local and global pruning, using L1 and L2 importance factors. We conduct a single iteration of pruning, varying the channel sparsity factor between 0.1 and 0.9 in increments of 0.1, and selecting the highest accuracy value for every 5% range total SubTuning params.

**Active Learning** In our experiments, we select examples according to their classification margin. At every iteration, after SubTuning our model on the labeled dataset, we compute the classification margin for any unlabeled example (similar to the method suggested in Balcan et al. (2007); Joshi et al. (2009a); Lewis & Gale (1994)). That is, given some example $x$, let $P(y|x)$ be the probability that the model assigns for the label $y$ when given the input $x$[5]. Denote by $y_1$ the label with the maximal probability and by $y_2$ the second-most probable label, namely $y_1 = \max_y P(y|x)$ and $y_2 = \max_{y \neq y_1} P(y|x)$. We define the classification margin of $x$ to be $P(y_1|x) - P(y_2|x)$, which captures how confident the model is in its prediction (the lower the classification margin, the less confident

---

[5]We focus on classification problems, where the model naturally outputs a probability for each label given the input. For other settings, other notions of margin may apply.

the model is). We select the examples that have the smallest classification margin (examples with high uncertainty) as the ones to be labeled.

In our Active Learning experiments we start with 100 randomly selected examples of the CIFAR-10 dataset. At each iteration we select and label additional examples, training with 500, 1000, 2500, 5000, and 10,000 labeled examples that were iteratively selected according to their margin. That is, after training on 100 examples, we choose the next 400 examples to be the ones closest to the margin, train a new model on the entire 500 examples, use the new model to select the next 500 examples, and so on. In Figure 11 we compare the performance of the model when trained on examples selected by our margin-based rule, compared to training on subsets of randomly selected examples. We also compare our method to full finetuning with and without margin-based selection of examples.

## B  ADDITIONAL EXPERIMENTS

### B.1  ADDITIONAL FINETUNING PROFILES

In this subsection we provide some more SubTuining profiles. We validate that longer training does not affect the ViT results, reporting the results in Figure 8. In Figure 9 we provide SubTuning results for 2 and 3 consecutive blocks. We show that using more blocks improves the classification accuracy, and makes the choice of later blocks more effective.

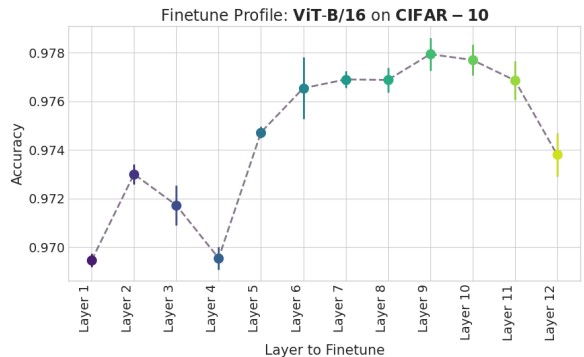

Figure 8: Finetuning profile trained with ViT-B/16 on Cifar10 trained for 40 epochs.

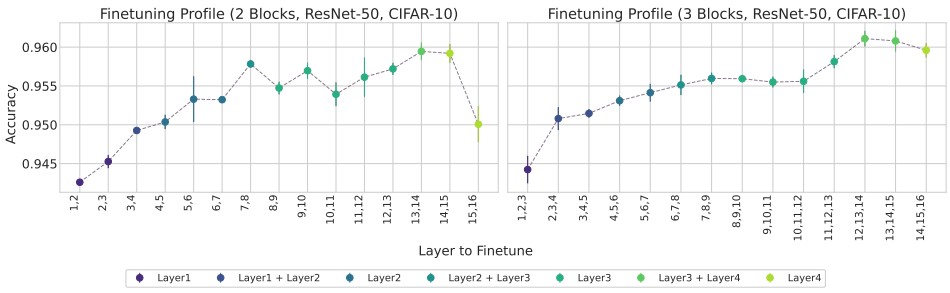

Figure 9: Finetuning profiles of ResNet-50 pretrained on ImageNet on CIFAR-10 with 2 blocks (*Left.*) and 3 blocks (*Right..*)

### B.2  ADDITIONAL PAIRWISE FINETUNING PROFILES.

We performed SubTuning with all pairs of residual blocks on the CIFAR-100, Flowers102, Caltech101 and DMLab 1k examples subsets from the VTAB-1k dataset. In addition, we also train on the entire CIFAR-100 dataset. We present our results in Figure10.

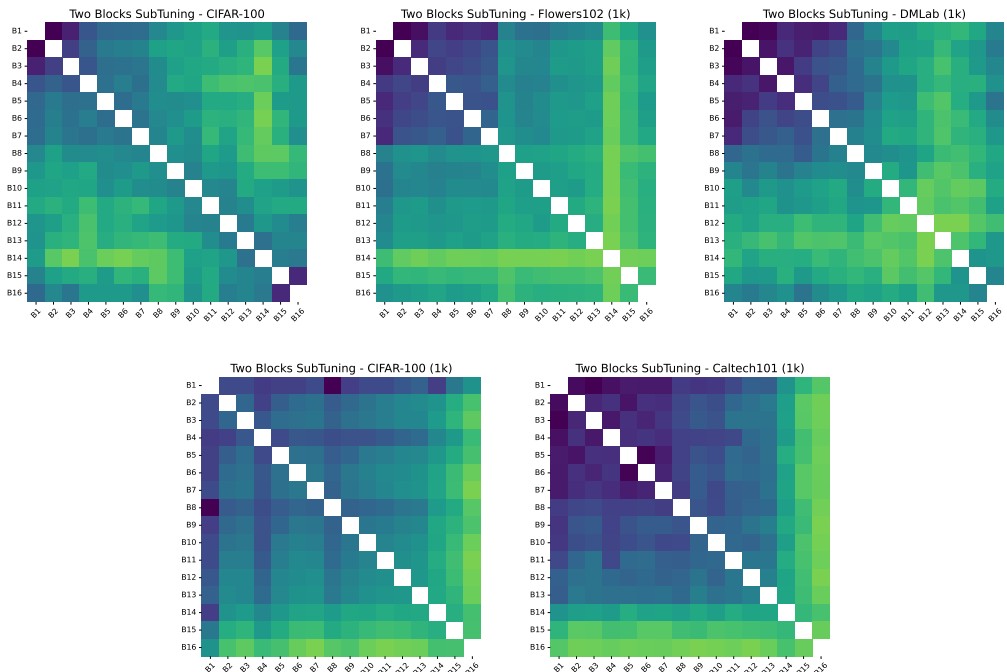

Figure 10: Two Blocks SubTuning of ResNet-50 on CIFAR-100, and the VTAB-1k versions of CIFAR-100, Flowers102, Caltech101 and DMLab datasets, denoted with 1k. We run a single experiment for any pair Blocks for 20 epochs.

Upon examining the outcomes for CIFAR-100 and DMLab datasets, it becomes evident that employing deeper Blocks yields superior performance when the dataset size is limited. However, for the DMLab dataset, utilizing SubTuning Blocks in the middle of the network appears to be more effective, despite the dataset's small size. This apparent inconsistency may be attributed to the unique characteristics of the dataset, which originates from simulated data, and the initial pretraining phase conducted on real-world data. These results underscore the importance of considering the specific properties of the dataset and the pretraining process when designing and optimizing the layer selection.

## B.3 ADDITIONAL DETAILS FOR SECTION 3

In table 5 we report the corresponding standard deviations for table 1 in the main body.

Table 5: Standard Deviation of ResNet-50 and ViT-b/16 pretrained on ImageNet and finetuned on datasets from VTAB-1k. FT denotes finetuning while LP stands for linear probing.

|  | ResNet50 | | | | ViT-b/16 | | | |
|---|---|---|---|---|---|---|---|---|
|  | CIFAR-100 | Flowers102 | Caltech101 | DMLAB | CIFAR-100 | Flowers102 | Caltech101 | DMLab |
| Ours | 0.0068 | 0.0056 | 0.0071 | 0.0064 | 0.029 | 0.0016 | 0.0076 | 0.0132 |
| H2T [6] Evci et al. (2022) | 0.14 | 0.08 | 0.25 | 0.13 | 0.29 | 0.5 | 0.16 | 0.14 |
| FT | 0.0085 | 0.0124 | 0.0206 | 0.01 | 0.0681 | 0.0226 | 0.0131 | 0.0132 |
| LP | 0.0051 | 0.0113 | 0.009 | 0.0054 | 0.0171 | 0.0079 | 0.0053 | 0.0028 |
| LoRA Hu et al. (2021) | - | - | - | - | 0.0348 | 0.0159 | 0.0147 | 0.0132 |

## B.4 EXTENSIONS AND ABLATIONS

In this section, we report additional results that we for SubTuning, that were omitted from or only partially discussed in the main body of the paper. Specifically, we study the interplay of SubTuning and Active Learning (Subsection B.4.1), how does reusing the frozen features affects performance (see Subsection B.4.2), the interaction between SubTuning and weight pruning (see Subsec-

tion B.4.3. and finally whether reinitilizing part of the weights can recover the finetuning performance (see Subsection B.4.4).

### B.4.1 ACTIVE LEARNING WITH SUBTUNING

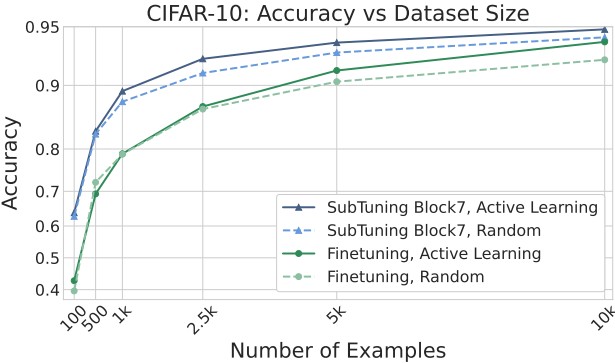

Figure 11: ResNet-50 pretrained on ImageNet with SubTuning on CIFAR-10 using Active Learning. We used logit scale for the y-axis to visualize the differences between multiple accuracy scales.

We saw that SubTuning is a superior method compared to both finetuning and linear probing when the amount of labeled data is limited. We now further explore the advantages of SubTuning in the pool-based Active Learning (AL) setting, where a large pool of unlabeled data is readily available, and additional examples can be labeled to improve the model's accuracy. It is essential to note that in real-world scenarios, labeling is a costly process, requiring domain expertise and a significant amount of manual effort. Therefore, it is crucial to identify the most informative examples to optimize the model's performance Katharopoulos & Fleuret (2018).

A common approach in this setting is to use the model's uncertainty to select the best examples Lewis & Gale (1994); Joshi et al. (2009a); Campbell et al. (2000); Joshi et al. (2009b). The process of labeling examples in AL involves iteratively training the model using all labeled data, and selecting the next set of examples to be labeled using the model. This process is repeated until the desired performance is achieved or the budget for labeling examples is exhausted.

In our experiments (see details in Appendix A.1), we select examples according to their classification margin. We start with 100 randomly selected examples from the CIFAR-10 dataset. At each iteration we select and label additional examples, training with 500 to 10,000 labeled examples that were iteratively selected according to their margin. For example, after training on the initial 100 randomly selected examples, we select the 400 examples with the lowest classification margin and reveal their labels. We then train on the 500 labeled examples we have, before selecting another 500 examples to label to reach 1k examples. In Figure 11 we compare the performance of the model when trained on examples selected by our margin-based rule, to training on subsets of randomly selected examples. We also compare our method to full finetuning with and without margin-based selection of examples. Evidently, we see that using SubTuning for AL outperforms full finetuning, and that the selection criterion we use gives significance boost in performance.

### B.4.2 SIAMESE SUBTUNING

In the multi-task setting discussed in Section 4, we have a network $f_\theta$ trained on one task, and we want to train another network by fine-tuning the weights $\theta$ for a different task, resulting in new weights $\widetilde{\theta}$. At inference time, we need to compute both $f_\theta(\mathbf{x})$ and $f_{\widetilde{\theta}}(\mathbf{x})$, minimize the additional cost of computing $f_{\widetilde{\theta}}(\mathbf{x})$, while preserving good performance. Since $f_\theta(\mathbf{x})$ is computed anyway, its features are available at no extra cost, and we can combine them with the new features. To achieve this, we concatenate the representations given by $f_\theta(\mathbf{x})$ and $f_{\widetilde{\theta}}(\mathbf{x})$ before inserting them into the classification head. This method is referred to as Siamese SubTuning (See illustration in Figure 12).

The effectiveness of Siamese SubTuning was evaluated on multiple datasets and found to be particularly beneficial in scenarios where data is limited. For instance, when finetuning on 5,000 randomly selected training samples from the CIFAR-10, CIFAR-100, and Stanford Cars datasets, Siamese

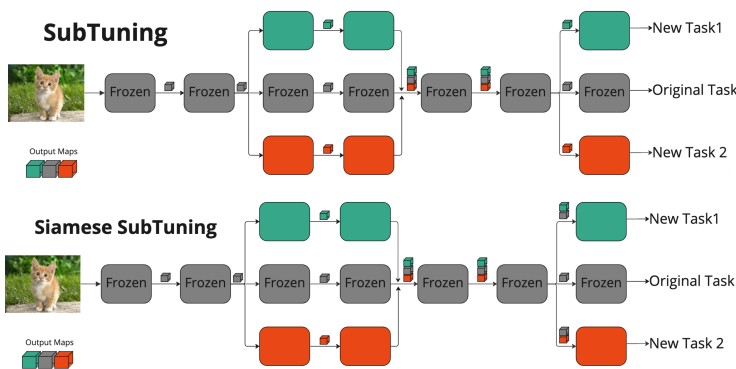

Figure 12: Illustration of SubTuning vs Siamese SubTuning. Note that the difference is that the new tasks now get the original features as input.

SubTuning with ResNet-18 outperforms standard SubTuning. Both SubTuning and Siamese Sub-Tuning significantly improve performance when compared to linear probing in this setting. For instance, linear probing on top of ResNet-18 on CIFAR-10 achieves 79% accuracy, where Siamese SubTuning achieves 88% accuracy in the same setting (See Figure 14).

Our comparison of SubTuning and Siamese SubTuning is based on experiments performed on 5,000 randomly selected training samples from CIFAR10, CIFAR100, and Stanford Cars datasets. Results for resblocks of ResNet-50 and ResNet-18 are provided in Figures 13 and 14 respectively. We also provide the results of full ResNet layers SubTuning, which involves SubTuning a few consecutive blocks that are applied to the same resolution (See Figure 15). As can be seen from the figures, Siamese SubTuning adds a performance boost in the vast majority of architectures, datasets, and block choices.

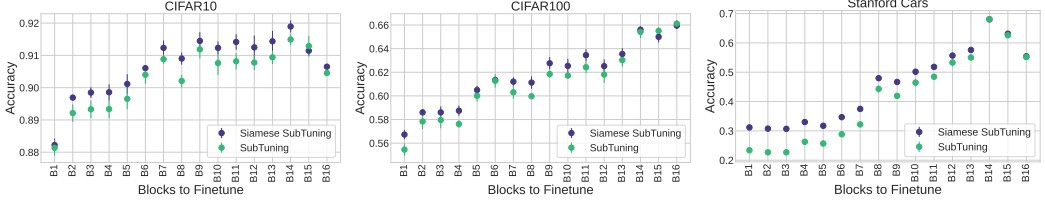

Figure 13: Siamese SubTuning on for ResNet-50 on CIFAR-10 (*left.*), CIFAR-100 (*middle.*) and Standford Cars (*right.*). We use 5,000 randomly selected training samples from each dataset.

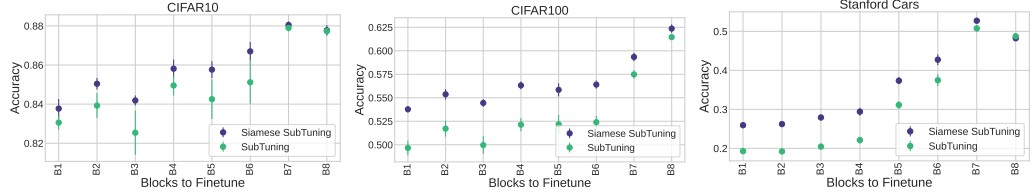

Figure 14: The impact of Siamese SubTuning on ResNet-18 when using 5,000 randomly selected training samples from each dataset.

### B.4.3 PRUNING

In our exploration of SubTuning, we have demonstrated its effectiveness in reducing the cost of adding new tasks for Multi-Task Learning (MTL) while maintaining high performance on those tasks. To further optimize computational efficiency and decrease the model size for new tasks, we introduce the concept of channel pruning on the SubTuned component of the model. We employ

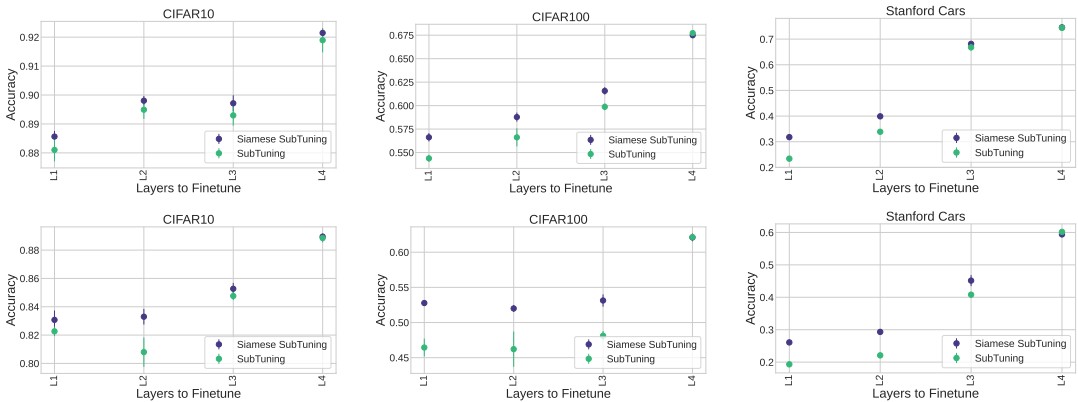

Figure 15: The impact of Siamese SubTuning of whole ResNet Layers (a group of blocks applied to the same resolution). Results for 5,000 randomly selected training samples from each dataset are presented for ResNet50 (**Top**) and ResNet18 (**Bottom**).

two types of pruning, local and global, to reduce the parameter size and runtime of the model while preserving its accuracy. Local pruning removes an equal portion of channels for each layer, while global pruning eliminates channels across the network regardless of how many channels are removed per layer. Following the approach of Li et al. Li et al. (2016), for both pruning techniques we prune the weights with the lowest L1 and L2 norms to meet the target pruning ratio.

The effectiveness of combining channel pruning with SubTuning on the last 3 blocks of ResNet-50 is demonstrated in our results. Instead of simply copying the weights and then training the blocks, we add an additional step of pruning before the training. This way, we only prune the original, frozen, network once for all future tasks. Our results show that pruning is effective across different parameter targets, reducing the cost with only minor performance degradation. For instance, when using less than 3% of the last 3 blocks (about 2% of all the parameters of ResNet-50), we maintain 94% accuracy on the CIFAR-10 dataset, compared to about 91% accuracy achieved by linear probing in the same setting.

All the pruning results for Local or Global pruning with L1 or L2 norms and varying channel sparsity factor between 0.1 and 0.9 in increments of 0.1 are presented in Figure 16. As we do not have a specific goal in performance or parameter ratio, we provide results for multiple fractions of the total SubTuning parameters and accuracy values. Despite slight differences between the methods, all of them yield good results in reducing the complexity of the SubTuning model with only minor accuracy degradation.

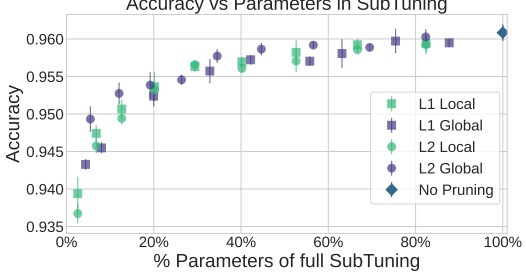

Figure 16: Full results of SubTuning with channel-wise pruning on the last 3 blocks of ResNet-50. We plot the accuracy vs the pruning rate of different pruning techniques (Global vs. Local and pruning norm) for different pruning rates.

### B.4.4 Effect of Random Re-initialization

In our exploration of SubTuning, we discovered that initializing the weights of the SubTuned block with pretrained weights from a different task significantly improves both the performance and speed of training. Specifically, we selected a block of ResNet-50, which was pretrained on ImageNet, and finetuned it on the CIFAR-10 dataset. We compared this approach to an alternative method where we randomly reinitialized the weights of the same block before finetuning it on the CIFAR-10 dataset. The results, presented in Figure 17, show that the pretrained weights led to faster convergence and better performance, especially when finetuning earlier layers. In contrast, random initialization of the block's weights resulted in poor performance, even with a longer training time of 80 epochs.

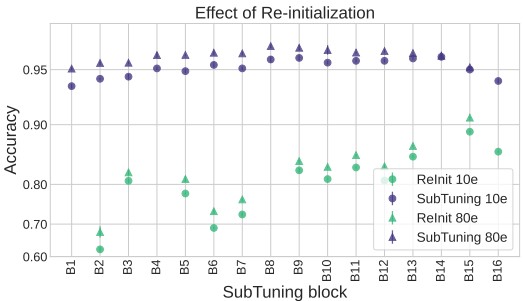

Figure 17: The effects of longer training and weight reinitializion on the *Finetuning Profile* of ResNet-50 pretrained on ImageNet and finetuned on CIFAR-10. For randomly re-initialized the weights, we encountered some optimization issues when training the first block on each resolution of the ResNet model, i.e. blocks 1, 4, 8 and 14. We use a logit scale for the y-axis, since it allows a clear view of the gap between different scales.

### B.5 Computational Efficiency

In this subsection we provide some more inference time results. In Figure 18 we provide the absolute results for SubTuning inference time for a different number of consecutive blocks. In Figure 19 we provide the accuracy vs the added inference time for 2 consecutive blocks. We can see that using blocks 13 and 14 yields excellent results both in running time and accuracy.

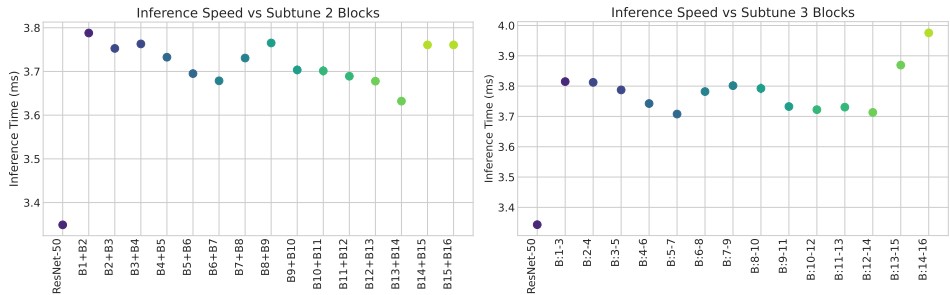

Figure 18: Absolute inference times for A100 GPU SubTuning on 2 and 3 blocks.

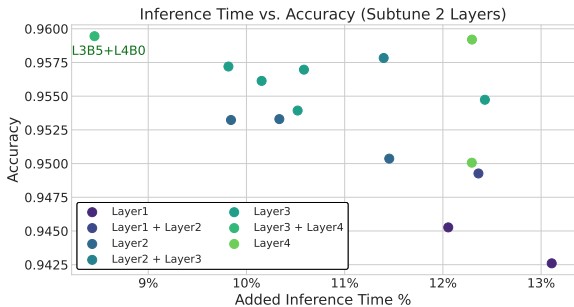

Figure 19: Accuracy vs inference time of two consecutive block SubTuning.

## C    PROOF OF THEOREM 1

We analyze a slightly modified version of the greedy SubTuning algorithm. For some set of pre-trained parameters $\theta$, and some subset of layers $S$, denote by $\theta_S$ the set of parameters of the layers in the subset $S$, and by $\psi_{\theta_S}$ the Neural Tangent Kernel (NTK) features induced by these parameters. We assume that for all $\mathbf{x}$ and $\theta$ we have $\|\psi_\theta(\mathbf{x})\|_\infty \leq 1$. For some $\mathbf{w}$, define the hypothesis $h_{\theta,S,\mathbf{w}}(\mathbf{x}) = \langle \psi_{\theta_S}(\mathbf{x}), \mathbf{w} \rangle$. Let $\ell$ be the hinge-loss, and denote the loss over the distribution $\mathcal{L}_\mathcal{D}(h) = \mathbb{E}_{(\mathbf{x},y)\sim\mathcal{D}}\left[\ell(h(\mathbf{x}), y)\right]$. Then, we define the algorithm that chooses the minimizer of the loss funciton over the NTK, subject to norm constraint $\Delta$:

$$\mathrm{evaluate}(\mathcal{D}, \theta, S, \Delta) = \min_{\|\mathbf{w}\|\leq\Delta} \mathcal{L}_\mathcal{D}(h_{\theta,S,\mathbf{w}})$$

We analyze the following algorithm:

---
**Algorithm 2** Greedy-SubTuning
---
1: **procedure** GREEDYSUBSETSELECTION(all_layers, $\mathcal{D}$, $\theta$, $\varepsilon$, $\Delta$, $r'$)
2:     $S \leftarrow \{\}$, $n \leftarrow |\mathrm{all\_layers}|$
3:     $A_{best} \leftarrow \mathrm{evaluate}(\mathcal{D}, \theta, S, \Delta)$
4:     **for** $i = 1$ to $n$ **do**
5:         $A_{iter} \leftarrow \infty$, $L_{best} \leftarrow$ null
6:         **for** $L \in (\mathrm{all\_layers} - S)$ **do**
7:             $S' \leftarrow S \cup \{L\}$
8:             $A_{new} \leftarrow \mathrm{evaluate}(\mathcal{D}, \theta, S', \Delta)$
9:             **if** $A_{new} < A_{iter} - \varepsilon$ **then**
10:                 $L_{best} \leftarrow L$, $A_{iter} \leftarrow A_{new}$
11:             **end if**
12:         **end for**
13:         **if** $A_{iter} < A_{best} - \varepsilon$ **and** $\mathrm{params}(S \cup \{L_{best}\}) \leq r'$ **then**
14:             $A_{best} \leftarrow A_{iter}$, $S \leftarrow S \cup \{L_{best}\}$
15:         **else**
16:             Break
17:         **end if**
18:     **end for**
19:     **return** $S$
20: **end procedure**
---

Fix some distribution of labeled examples $\mathcal{D}$. Let $\mathcal{S}$ be a sample of $m$ i.i.d. examples from $\mathcal{D}$, and denote by $\widehat{\mathcal{D}}$ the empirical distribution of randomly selecting an example from $\mathcal{S}$. Fix some $\delta'$. Then, using Theorem 26.12 from Shalev-Shwartz & Ben-David (2014), for every subset of layers $S$ with at most $r'$ parameters, with probability at least $1 - \delta'$, for every $\mathbf{w}$ with $\|\mathbf{w}\| \leq \Delta$:

$$\left|\mathcal{L}_\mathcal{D}(h_{\theta,S,\mathbf{w}}) - \mathcal{L}_{\widehat{\mathcal{D}}}(h_{\theta,S,\mathbf{w}})\right| \leq \frac{2\sqrt{r'}\Delta}{\sqrt{m}} + \left(1 + \sqrt{r'}\Delta\right)\sqrt{\frac{2\log(4/\delta')}{m}}$$

Now, let $S_1, \ldots, S_T$ be all the subsets that are being evaluated during the runtime of GreedySubsetSelection(all_layers, $\mathcal{D}, \theta, \varepsilon, \Delta, r'$), namely the algorithm running on the true distribution $\mathcal{D}$. Note that if there are $n$ layers in the model, then there are at most $n^2$ such subsets. Let

$$m > \frac{16r'\Delta^2}{\varepsilon^2} + 2\left(1 + \sqrt{r'}\Delta\right)^2 \frac{2\log(4n^2/\delta)}{\varepsilon^2} = O\left(\frac{r'\Delta^2\log(n/\delta)}{\varepsilon^2}\right)$$

using the union bound we get that, w.p. at least $1 - \delta$, for all $S_1, \ldots, S_T$ it holds that $\left|\mathcal{L}_{\mathcal{D}}(h_{\theta,S_i,\mathbf{w}}) - \mathcal{L}_{\widehat{\mathcal{D}}}(h_{\theta,S_i,\mathbf{w}})\right| \leq \varepsilon/2$. Therefore, we have that $\left|\text{evaluate}(\mathcal{D}, \theta, S_i, \Delta) - \text{evaluate}(\widehat{\mathcal{D}}, \theta, S_i, \Delta)\right| \leq \varepsilon/2$ for all $S_i$. This means that, with probability at least $1 - \delta$, running GreedySubsetSelection(all_layers, $\widehat{\mathcal{D}}, \theta, \varepsilon, \Delta, r'$) must choose the subsets $S_1, \ldots, S_T$. Since we showed that $\left|\mathcal{L}_{\mathcal{D}}(h_{\theta,S_T,\mathbf{w}}) - \mathcal{L}_{\widehat{\mathcal{D}}}(h_{\theta,S_T,\mathbf{w}})\right| \leq \varepsilon/2$ we get the required generalization guarantee on the output of the empirical algorithm.

