# OpenReview forum: "Less is More: Selective Layer Finetuning with SubTuning"
_ICLR.cc/2024/Conference — Submitted to ICLR 2024_

### Official Review · Reviewer_rTPG · 2023-10-17

**Soundness:** 2 fair
**Presentation:** 3 good
**Contribution:** 3 good
**Rating:** 3
**Confidence:** 4

**Summary:**

The paper works with parameter efficient finetuning/transfer learning. Specifically, given a model pretrained on some source task, and later to be adapted for some downstream task, the paper discusses the phenomenon that selectively fine-tuning a subset of layers often does better than fine-tuning the linear head or the entire network, specially if there is data scarcity. Moreover, this subset of layers is not predetermined, and rather depends on what the distribution shift between source and downstream task is. The authors introduce a greedy algorithm of choosing the best subset of layers for a particular task, and shows good performance on a variety of vision datasets.

**Strengths:**

1. The paper’s studied problem is well-motivated.
2. I liked the idea of the greedy algorithm they use to choose the subset of layers to fine-tune: I found this algorithm to be simple yet effective. I also thought the algorithm is novel.
3. The paper is well-organized and easy to follow, and have comprehensive experiments.

**Weaknesses:**

**Weakness 1 (lack of discussing prior work)**

The paper does not mention a lot other prior work that looked into fine-tuning particular layers for specific purposes. For example:
1. DFR [1] is an important work that studies retraining only the last layer to get rid of spurious correlations.
2. The paper discussed linear probing and fine-tuning, but does not discuss LP-FT [2]. There is a complex dynamic behind LP and FT, specially when distribution shift is involved. It is also important to consider pre-trained feature distortion, which can be a reason sub-tuning works better than full fine-tuning.

Finally, the paper’s claim/observation in the abstract is not novel, i.e., it has been recorded in other papers before [7], and should be modified as such.

> **This paper (abstract)**: We observe that not all layers are created equal: different layers across the network contribute variably to the overall performance, and the optimal choice of layers is contingent upon the downstream task and the underlying data distribution

> **Surgical fine-tuning [7] (abstract)**: This paper shows that in such settings, selectively fine-tuning a subset of layers (which we term surgical fine-tuning) matches or outperforms commonly used fine-tuning approaches. Moreover, the type of distribution shift influences which subset is more effective to tune.


**Weakness 2 (formal definition/terminology):**

Setting up a formal definition of the problem would make the paper much easier to read. For example, one could define it as: let $S$ be the set of all layers, then sub-tuning is:
$$S_{best} = argmin_{S' \subseteq S} L(f, S', D_{ft}, D_{val})$$

where $L(f, S', D_{ft}, D_{val})$ is the loss after fine-tuning the network $f$'s layers in $S'$ on dataset $D_{ft}$ and evaluating the final performance on $D_{val}$.

With this in mind, how is sub-tuning different than surgical fine-tuning? I think the main difference is not in the formal definition, but in the practical implementation: surgical fine-tuning only considers $S' \subseteq S$ that are contiguous, whereas this paper considers $S'$ that are not necessarily contiguous. If this is the case, this should be proposed as an improved/better way of doing surgical fine-tuning, not a separate problem definition. If not, please explain.

Also a proper explanation of what authors consider as blocks vs layers in a neural network would be important. Ideally, some sort of schematic of ResNet-50, discussing the terminology, in the main paper/appendix would be needed.

**Weakness 3 (experiment results):**
1. **Figure 1**: The difference between fine-tuning **B1** (94%) vs **B14** (95.5%) in figure 1 (left) is negligible. Also denoting the number of parameters in each of these layers would be important.
2. **Figure 3**: I would interpret the results in a different way: looking at the grid column wise, it seems the columns corresponding to B13, B14, B15 do well in general. I would also guess that the variation of accuracy within these columns is small when adding B2, B4, B8 or B9. How much is adding the second block possibly helping here? It can quite possibly be that fine-tuning B13/14/15 already recovers most of the performance/does most of the heavy lifting, and the improvement that comes from fine-tuning B2 in addition to B14 is negligible.
3. **See question 3.** I seriously doubt that in the limit of infinite data, subtuning essentially chooses close to all layers, and therefore becomes full fine-tuning effectively. If that is not the case, I am curious why it is not? Given infinite data, and more layers to fine-tune, the neural network should intuitively learn a better classifier, unless only a few layers can solve the problem effectively/the layers not fine-tuned are already close to optimum for the downstream task. For $\textrm{source} \rightarrow \textrm{target}$ dataset with large distribution shift, this should never be the case, so I suspect this result is because of $\textrm{Imagenet} \rightarrow \textrm{CIFAR-10}$ shift. If this is not the case, please explain.
4. **Table 1**: Missing some very important comparisons, like LP-FT [2], layer-wise learning rate tuning [6], gradual unfreezing [3, 4, 5].
5. **Section 3.2**: Missing results on some very important benchmark. The section title contains distribution shift, but focuses solely on image corruptions. Some results on natural shifts such as the WILDS [8] (i.e., at least 2 or 3 of these datasets) would strengthen the paper.
6. **Section 3.2**: “Furthermore, our results challenge the claim made in Lee et al. (2022b) that suggests adjusting only the first layers of the network suffices for input-level shifts in CIFAR-10-C. Interestingly, we found that the ultimate or penultimate block was the first layer selected for all corruptions, resulting in the largest performance increase.” I feel this claim is not well made. For example, even in the reproduced results of this paper, we see surgical L1 > surgical L2 > surgical L3. Note that sub-tuning looks at a much more granular level (i.e., within blocks of a single layer), and finds a better finetuning arrangement. Also, it is possible that the finetuning the last layer performs worse, but the last block within the last layer performs better, hence this is not a contradiction. Furthermore, in figure 5, B12 is the most commonly chosen first block, but B1 is the most commonly chosen second block, so the first block is still important? In what order would sub-tuning pick the parameters, if allowed only to fine-tune layers and not the blocks inside?

**Weakness 4 (lack of interpretability):**

The authors give a mechanistic way of choosing the subset of layers that needs to be fine-tuned, but there is no understanding of the layers chosen other than it empirically works well. Why would, for a particular downstream task, layers B8 and B14 combine to do the best? Possible things to explore here:
1. How much does adding the second layer help? How much does adding the third layer? And so on.
2. Possibly looking at something like [9]. It is hard to do for vision experiments, but understanding why certain combinations of layers help is important.

Otherwise, this method is computationally heavy (requires multiple fine-tuning rounds) and may not be suitable for use in practice compared to full fine-tuning, LP-FT or surgical fine-tuning for their simplicity.

**Questions:**

1. **Finetuning profiles (figure 1 and figure 2)**: how many data points are used to fine-tune each of the layers?
2. **Page 5, greedy selection**: “We note that such greedy optimization is a common approach for subset selection in various combinatorial problems, and is known to approximate the optimal solution under certain assumptions.” Explain the assumptions, and/or add references for this.
3. **Page 5, greedy selection**: “We show that SubTuning results in comparable performance to full finetuning even for full datasets”. Does this hold only for CIFAR-10? Or does this hold for other datasets as well? Also how many layers are being fine-tuned when the full-dataset is used? I.e., does subtuning essentially become full fine-tuning in the limit of the full dataset?
4. **Table 1**: Why would surgical fine-tuning [7] not be applicable for ViT-B/16 architecture? Table 1 in [7] shows surgical fine-tuning using different attention blocks on a ViT-B/16 architecture. One doesn’t need to group attention blocks into layers, one can just choose one/two/three contiguous attention blocks and fine-tune them for surgical FT.
5. **Figure 4**: Does the authors have any intuition for the results in figure 4? Why would sub-tuning pick earlier layers, i.e., layers closer to the input, if more data is available?
6. **Computational cost of sub-tuning**: What is the cost of sub-tuning? The big-Oh complexity is given in the paper, but sub-tuning might occasionally finish earlier before checking all blocks. Could the authors provide a table of computational cost, either in terms of total fine-tuning epochs required, summed over checking each block set until their algorithm terminates, or FLOPs or some other measure, between just running full finetuning once, running surgical fine-tuning and sub-tuning?
7. **Active learning**: What does classification margin mean here? Is this simply the maximum softmax/prediction probability?
8. **Figure 6, Subtuning for MTL**: I am not entirely sure how this, specially the split in the computation graph in figure 6, works. Say for a new task, you only need to tune blocks B5 and B6. Would you initialize another copy of these blocks, B5’ and B6’, with the same pretrained weights as B5 and B6, and for the new task, you only update these copies of the weights B5’ and B6’, and leave the original blocks/weights B5 and B6 unchanged? Keeping a second copy of these blocks with original pretrained weights would prevent catastrophic forgetting. I am not entirely sure how this is done.
9. **Siamese subtuning**: For Siamese subtuning, do you have to double the number of parameters in the classification head? I.e., consider the classification head of the original task (say 512 dimensions) and **only** the new task (say another 512 dimensions) for regular subtuning. Then for siamese subtuning, since the new task head receives both embeddings for original and new task, do you update the new task classifier head to be 512 dimensional? Also when you finetune on a new task, do you prevent gradient updates due to the original task’s embedding going to a new task’s classification head? Or do you allow the new task to change the original task’s weights?
10. **$\mathbf{\epsilon}$**: What is the threshold, $\epsilon$, in algorithm 1? This is a very important hyper-parameter, and I did not find any discussion on how this is picked. Also my guess is it is pretty small, since the performance gain keeps decreasing (in trend) as one adds more layers.
11. **$\mathbf{\epsilon}$**: Could you add results on how robust the algorithm is to these chosen threshold, $\epsilon$?
12. **Optimizer**: What is the optimizer used for training ViT-B/16?


[1] Last Layer Re-Training is Sufficient for Robustness to Spurious Correlations, https://arxiv.org/abs/2204.02937

[2] Fine-Tuning can Distort Pretrained Features and Underperform Out-of-Distribution, https://arxiv.org/abs/2202.10054

[3] Universal Language Model Fine-tuning for Text Classification, https://arxiv.org/abs/1801.06146

[4] Distilling BERT into Simple Neural Networks with Unlabeled Transfer Data, https://arxiv.org/abs/1910.01769

[5] Targeted transfer learning to improve performance in small medical physics datasets, https://arxiv.org/abs/1912.06761

[6] AutoLR: Layer-wise Pruning and Auto-tuning of Learning Rates in Fine-tuning of Deep Networks, https://arxiv.org/abs/2002.06048

[7] Surgical Fine-Tuning Improves Adaptation to Distribution Shifts, https://arxiv.org/abs/2210.11466

[8] WILDS: A Benchmark of in-the-Wild Distribution Shifts, https://arxiv.org/abs/2012.07421

[9] Locating and Editing Factual Associations in GPT, https://arxiv.org/abs/2202.05262

---

### Official Review · Reviewer_nACT · 2023-10-29

**Soundness:** 3 good
**Presentation:** 3 good
**Contribution:** 2 fair
**Rating:** 5
**Confidence:** 4

**Summary:**

This paper proposes a new parameter-efficient fine-tuning method, i.e., subtuning to improve the model’s transfer learning performance. The proposed method is experimentally proven to perform better than other fine-tuning baselines. The paper is easy to follow, and the experiments are diverse and clear enough to show the advantages of the proposed method. However, it is a little hard for me to understand why this method works based on the context of the current version. Hence I tend to give a 5 at this stage, and I am happy to increase my evaluation in the rebuttal phase. Because I believe the method proposed in this paper is simple and effective in many practical cases.

**Strengths:**

1. The method is simple and easy to understand. The performance gain is good.
2. The concept of “finetuning profiles”, which demonstrates the contribution of different layers to the final results, is interesting. (But I believe only showing them is not enough. Explaining why the profiles behave like this under specific scenarios can make the paper stronger.)
3. Abundant experimental results from different perspectives.

**Weaknesses:**

1. The theoretical guarantee and explanations are not enough. Although theorem 1 provides a generalization bound that depends on the number of stunned parameters, it is hard to link this theory to the algorithm studied in this paper. Maybe the analysis provided in [1, 2] (the analysis on an overparameterized model) would be helpful.
2. The results in Table 1 are based on VTAB-1k, which means the samples used for transfer learning are only 1k. Will the proposed method still work when fine-tuning using the full CIFAR100 dataset? AFAIK, fine-tuning a pretrained ResNet50 on full CIFAR100 should achieve roughly 70%+ accuracy, which is much higher than the numbers provided in the paper. It is easier to compare with other baselines using similar settings.
3. It would be very helpful if the paper could briefly introduce how the VTAB-1k dataset is generated (maybe in the appendix).
4. In Section 4 and Figure 6, the paper claims that subtuning can bring benefits to multi-task learning. However, I cannot find the corresponding experimental results or analysis supporting that.

**Questions:**

1. The paper compares the proposed subtuning with the basic fine-tuning and linear probing methods, which is good. However, [1, 2] analyze why linear probing and fine-tuning alone are not the best choice for transfer learning tasks. They propose to first linear probe several epochs and then finetune the whole network. I believe this method should also be considered as one of the baselines.
2. Following the previous point, the subtuning method can also be combined with fine-tuning. Similar to the method proposed in [2], it might be interesting to see whether the performance can be further improved if we finetune the whole network together after subtuning.

[1] Kumar, Ananya, et al. "Fine-tuning can distort pretrained features and underperform out-of-distribution." ICLR-2022.

[2] Ren, Yi, et al. "How to prepare your task head for finetuning." ICLR-2023

---

### Official Review · Reviewer_zhTK · 2023-10-29

**Soundness:** 2 fair
**Presentation:** 3 good
**Contribution:** 1 poor
**Rating:** 3
**Confidence:** 4

**Summary:**

The paper presents a parameter-efficient fine-tuning method, referred to as SubTuning, which greedily selects and fine-tunes a subset of layers in a pretrained model. It evaluates the performance of SubTuning on several vision models (e.g., ResNet50, ViT) using datasets like CIFAR-100, Flowers102, and Caltech101. Empirical results indicate that the proposed method outperforms naive baselines (e.g., linear probing and full finetuning) in low-data/distribution-shift settings.

**Strengths:**

1. The overall problem of fine-tuning only the necessary layers to save compute is interesting.
2. There are some valuable empirical observations made by the paper, such as the fine-tuning profiles and the layer importance of different network architectures.
3. The presentation is clear, and the idea is easy to follow. The visualization also helps understanding the method's performance.

**Weaknesses:**

1. **Lack of novelty:** Performing layer selection during fine-tuning has been explored by previous work [1, 2, 3]. These works use more advanced selection techniques like policy networks or genetic algorithm. The proposed way of measuring layer importance through simply fine-tuning accuracy on a specific dataset is also not generalizable, i.e., the observed trend only holds for a specific model and a specific dataset and it is hard to deduce any more general and useful rules from that.

2. **Related work:** The paper briefly mentions related work in parameter-efficient transfer learning and multitask learning. However, the idea of layer selection is also very relevant to layer-wise adaptive learning rate [4] (since a learning rate of 0 is the same as freezing the layers) and neural architecture search for parameter-efficient transfer learning [5]. The idea of layer importance is relevant to the neural network pruning literature as well [6]. A more thorough discussion and comparison with these methods would provide valuable context and insights.

3. **Limited models and datasets:** The paper only evaluates a few variants of ResNet and ViT, which is relatively small in scale compared to state-of-the-art large vision models. It is not sure whether the proposed method can scale up to these more practical models. Also, only 4 relatively simple datasets are used for evaluation (CIFAR, Flowers102, Caltech101, DMLAB). It would be better to see how the proposed method performs on more datasets with larger distribution shifts.

[1] SpotTune: Transfer Learning through Adaptive Fine-tuning. Guo et al. 2019.

[2] Dynamic fine-tuning layer selection using Kullback–Leibler divergence. Wanjiku et al. 2022.

[3] Automatic layer selection for transfer learning and quantitative evaluation of layer effectiveness. Nagae et al. 2022.

[4] Not All Layers Are Equal: A Layer-Wise Adaptive Approach Toward Large-Scale DNN Training. Ko et al. 2022.

[5] Neural Architecture Search for Parameter-Efficient Fine-tuning of Large Pre-trained Language Models. Lawton et al. 2023.

[6] Layer-adaptive Sparsity for the Magnitude-based Pruning. Lee et al. 2021.

**Questions:**

1. In Table 1, does the "FT" baseline means full fine-tuning? If so, why is the performance so bad (even worse than linear probing)?

---

### Official Review · Reviewer_NuCG · 2023-11-01

**Soundness:** 2 fair
**Presentation:** 2 fair
**Contribution:** 2 fair
**Rating:** 5
**Confidence:** 4

**Summary:**

The main contribution of the paper is the observation that finetuning only a subset of layers can outperform finetuning all the parameters. This subset of layers is selected using a greedy algorithm called SubTuning which relies on finetuning profiles of different layers to greedily pick the best performing layer for a downstream task. The paper shows how SubTuning can be particularly helpful when training data is scarce. The paper then shows how SubTuning, despite being expensive during training, can lead to efficient inference especially in the multi-task setting.

**Strengths:**

1. The proposed method can be applied in conjunction with other parameter-efficient methods such as LoRA and Head2Toe.
2. The method is simple and can be broadly applied to any architecture / domain.
3. Some observations in Section 2 are quite interesting, for example how the finetuning profiles change for different downstream tasks and for different pre-training methods (even for the same architecture) is quite intriguing to me. This indicates that the kind of features learned by different pre-training methods can be very different and can have vaying levels of usefulness even for the same downstream task.
4. The results on corrections and distribution shifts are quite promising.
4. The initial observations about efficiency at inference time in Section 4 are also interesting (though I have some concerns, see weaknesses below).

**Weaknesses:**

1. I have some concerns about the contributions of this paper -- the key idea that we do not need to finetune all layers and that finetuning only a subset of the parameters can outperform full-feature finetuning was already observed in the "Surgical finetuning" paper of Lee et al. To me, the only new thing shown in this paper (beyond what was already shown in Lee et al), is the fact that we can greedily choose more than just one block of layers to finetune. In Related Work, the authors mention "While our work shares some similarities with Lee et al., the motivations and experimental settings are fundamentally different" -- I'm a little confused by this sentence. As I understand, the experimental settings are exactly the same, and this is also mentioned in the paper later in the data corruption section where the authors say "...we follow the setup proposed in Lee et al". In related work the authors also mention "..we delve deeper into the complex interaction between the appropriate layers to finetune and the downstream task, pretraining objective, and model architecture, and observe that a more nuanced viewpoint is required" -- I'm again very confused by this, what exactly is the nuance and how does it add to what we already know from Lee et al? It'd be very useful if the authors can precisely state their contributions and mention what they add beyond the insights in Lee et al.

2. I like the observations in Section 2 and would've liked a more thorough analysis of how finetuning profiles change with architecture, pretraining, and downstream tasks. Right now I only see (from the first row in Fig 2) how things change on CIFAR10 with different architectures. But then again I notice that all these architectures don't have the same pretraining (ViT is on ImageNet-21k, Resnet18 is ImageNet-1k and ResNet50 is pre-trained using DINO). It might make sense to do this more systematically, one where you hold the pretraining the same and vary architectures and downstream tasks. One where you hold an architecture constant and show different pre-training methods. If this is one part of the "nuance" you add in your paper beyond Lee et al's work, then it requires a proper investigation.

3. Under "The Finetuning Profile" in Section 2, it might help to precisely define what you're profiling for (ie which accuracy are you reporting), only by reading until Section 3.1 I realized that this is cross-validation accuracy.

4. The observation in Section 2 ("Not All Layers Are Created Equal") has been made in prior work as well [1], it's worth citing this paper.

5. In Section 2 where you discuss "The Finetuning Profile", you've already made the assumption that the right unit at which to finetune is a block of layers, can you elaborate on why this was done? Why not layers within a block or even portions of layers?

6. Under "Results" in Section 2 you mention that it's surprising that layers with fewer parameters can outperform layers with more parameters. However, I do not quite understand why # parameters are more relevant than the quality of features? In fact, your own results show that B16 and B8 when finetuned together (Fig 3) outperform the case when either of these blocks is finetuned alone, indicating parameter count probably has nothing to do with the finetuning profile, and that what (probably) matters more is the nature of features being captures by a block. Am I missing something?

7. I'm not sure what the theoretical analysis in Section 2.1 adds to the paper. I again do not understand why parameter count (r) matters more than the quality of features or inductive bias. For example, based on the error bound in Theorem 1, wouldn't you always choose k=1 and choose this such that you minimize r (hence you'll always choose the layer with the least number of params)? But this is obviously not true based on your empirical results. Am I interpreting the theorem incorrectly?

8. Section 4 starts by discussing how SubTuning can be efficient for a multi-task setup, however, all experiments shown in Section 4.1 do not consider any multi-task setup. Additionally, section 4 starts by saying you will focus on FLOPS for compute and bits needed in memory for IO considerations. However, the results in Figure 7 show Inference Time on the x-axis. Why is this the case?

9. I like where Section 4 is going but I feel like it requires a more succinct argument and more convincing experiments. For example, to me it's not clear how you will save on FLOPS -- the moment you choose layer (or block) 0 as a finetuning block in SubTuning then # FLOPS = same as the FLOPS needed with full finetuning.

[1] Are All Layers Created Equal? https://arxiv.org/abs/1902.01996

**Questions:**

1. In Section 3.1 under VTAB-1k you mention"...using the 1k examples split specified in the protocol". What is this referring to? I could not find any details on the splits of these datasets.
2. What are the stddevs in Table 1?
3. You mention "we use the official PyTorch ResNet50" -- which weights are these? There are two weights in the official PyTorch repo, V1 and V2 (https://pytorch.org/vision/stable/models.html).
4. Can you increase the size of Figs 4 and 3, they are barely legible currently.

With some more work, I think this has the potential to be a cool paper!

---

### Meta-Review · Area_Chair_GZVV · 2023-12-06

**Metareview:**

The paper proposes SubTuning, a method for finetuning a pretrained model while freezing the majority of the layers. Specifically, authors propose a greedy algorithm for finding an optimal subset of layers to finetune, and only finetune those layers. The authors show promising results on finetuning to various tasks, including distribution shift tasks.

## Strengths

The proposed method is interesting and makes intuitive sense. The empirical results are promising. The authors evaluate the method on a range of problems.

## Weaknesses

Multiple reviewers pointed out the similarity of this work to [_Surgical Fine-Tuning Improves Adaptation to Distribution Shifts_](https://arxiv.org/abs/2210.11466). The main difference between the two papers seems to be that SubTuning can select non-consecutive layers. Reviewers also pointed out other similar papers.

The reviewers also highlighted that there isn't sufficient understanding of why the method works. It is unclear intuitively why it is sensible to only finetune layer 8 and layer 14.

**Justification For Why Not Higher Score:**

The authors did not provide a rebuttal to address the concerns of reviewers. The reviewers voted for rejection and raised constructive criticism.

**Justification For Why Not Lower Score:**

N/A

---

### Decision · Program_Chairs · 2024-01-16

Reject